# Impact persistence of stock market risks in commodity markets: Evidence from China

**Shusheng Ding[1], Zhipan Yuan[2], Fan Chen[1], Xihan Xiong[3], Zheng Lu[4], Tianxiang Cui[4]***

**1** School of Business, Ningbo University, Ningbo, Zhejiang, China, **2** School of Economics and Management, Northeast Agricultural University, Harbin, Heilongjiang, China, **3** Department of Computing, Imperial College London, London, United Kingdom, **4** School of Computer Science, University of Nottingham Ningbo China, Ningbo, Zhejiang, China

* tianxiang.cui@nottingham.edu.cn

**Data Availability Statement:** The data is downloaded from the WIND database (https://www.wind.com.cn/en/edb.html). The dataset is extracted from the commodity section of the WIND database for the Chinese commodity markets, ranging from January 1, 2010 to March 22, 2021.

## Abstract

The risk spillover among financial markets has been noticeably investigated in a burgeoning number of literature. Given those doctrines, we scrutinize the impact persistence of volatility spillover and illiquidity spillover of Chinese commodity markets in this paper. Based on the sample from 2010 to 2020, we reveal that there is a cross-market spillover of volatility and illiquidity in China and also, interactions between volatility and illiquidity in different financial markets are pronounced. More importantly, we demonstrate that different commodity markets have different responsiveness to stock market shocks, which embeds their market characteristics. Specifically, we discover that the majority of the traders in gold market might be hedger and therefore gold market is more sensitive to stock market illiquidity shock and thus the shock impact in persistent. On the other hand, agricultural markets like corn and soybean markets might be dominated by investors and thus those markets respond to the stock market volatility shocks and the shock impact in persistent over 10 periods given the first period of risk shock happening. In fact, different Chinese commodity markets' responsiveness towards Chinese stock market risk shocks indicates the stock market risk impact persistence in Chinese commodity markets. This result can help policymakers to understand the policy propagation effect according to this risk spillover channel and risk impact persistence mechanism in China.

## 1 Introduction

April 20, 2020, was a memorable day for the oil futures market in the US. It was the first time in history that the US oil futures price exhibited a negative price on that day. The May contract for crude-oil benchmark-grade West Texas Intermediate (WTI) plummeted to -$37.63 within minutes (see Fig 1). Afterwards, the US Energy Information Agency (EIA) provided two reasons for this historic event: lack of market liquidity because the May contract was near maturity and limited available storage for crude oil [1].

From the above case, it can be seen that market liquidity plays a crucial role in commodity futures trading. As a result, the illiquidity risk spillover, may have a substantial effect on

The other data is extracted from the stock market section of the WIND database for the CSI 300 index data, ranging from January 1, 2010 to March 22, 2021.

**Funding:** The author(s) received no specific funding for this work.

**Competing interests:** The authors have declared that no competing interests exist.

trading activities in commodity markets. Andrikopoulos et al. [2] maintained that information and market sentiment transmissions could be realized through illiquidity spillover across different financial markets. Smimou and Khallouli [3] show illiquidity co-movement and propagation in country-level financial markets inside the Eurozone. More importantly, Cespa and Foucault [4] demonstrate that illiquidity spillover could create market fragility, where a small reduction in market illiquidity could be exacerbated into a large market liquidity crash through a feedback loop. More recently, Amihud and Noh [5] substantiate that market illiquidity exhibits considerable effect in both cross-section and time-series dimensions in the stock market.

On the other hand, the strong link between market illiquidity and volatility has been substantially examined in contemporaneous studies (see Bao and Pan [6]; Valenzuela et al. [7]; Iwatsubo et al., [8]). Indeed, the fact that volatility is a key determinant of market liquidity is also well documented (see Chan et al. [9]; Chung and Chuwonganant [10]). The crude oil market plays an important part in commodity markets, and thus, crude oil market volatility spillover has been substantially investigated by the existing literature. Many studies scrutinize the volatility transmission between the oil market and agricultural markets (see Du et al., [11]; Nazlioglu et al., [12]; Kang et al., [13]) as well as the stock market (Arouri et al., [14]; Khalfaoui et al., [15]; Xu et al., [16]).

Since illiquidity spillover is a critical issue in financial markets and volatility is a vital determinant of market liquidity, the key contribution of this paper is to study the impact persistence of illiquidity spillover and volatility spillover among different commodity markets. We intend to explore the impact sequence between volatility spillover and illiquidity spillover, which enables us to analyze whether volatility spillover results in further illiquidity spillover. We mainly focus on the stock market risk spillover towards the commodity markets. This argument is based on volatility serving as the determinant of market liquidity and, more importantly, the negative relation between volatility and market liquidity, which has been discussed in plenty of studies (see Chan et al. [9]; Barinov [17]; Chung and Chuwonganant [18]).

Recently, massive studies have been devoted to exploring the relationship between commodity market risk spillovers with other market shocks. Ahmed and Huo [19] investigate the dynamic relationship among the stock market, commodity markets and global oil price. They reveal the significant unidirectional return spillover effect from oil market to stock market in

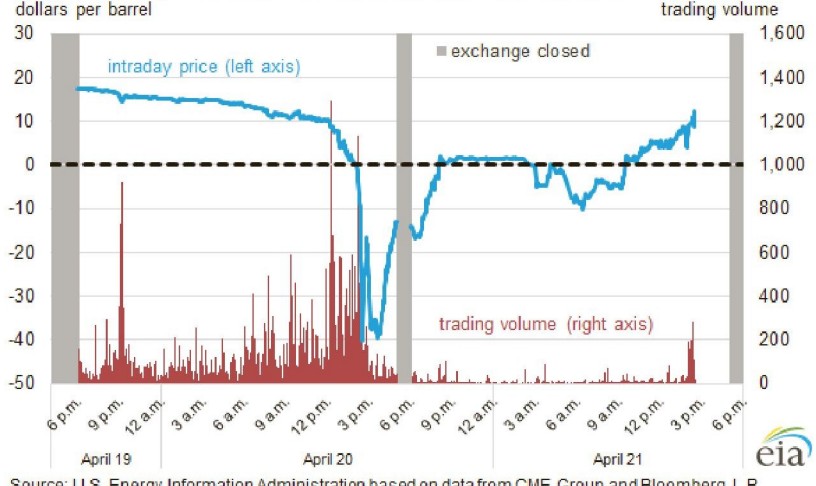

**Fig 1. May 2020 West Texas Intermediate futures contract pricing movement on April 20, 2020.**

China by applying a trivariate VAR-BEKK-GARCH model. More strikingly, Frommel et al. [20] unveil that low liquidity-beta stocks outperform high liquidity-beta stocks on a risk-adjusted basis during 1997 and 2016 in Chinese financial markets. They propose a competing behavioral-based explanation to explain this liquidity beta anomaly in China. Additionally, Umar et al. [21] examine the interdependence between oil price volatilities and agricultural commodities from 2002 to 2020 by employing Granger causality tests. They apply daily data from Coronavirus Worldwide Panic Index and commodity prices to conclude that oil shocks can be used to predict the future agricultural commodities and vice versa. Zhang and Ding [22] also attempt to scrutinize the liquidity effect on commodity prices and return movements. Their research demonstrates the liquidity has an inferencing effect on the commodity price co-integration. In the more recent work of Umar et al. [23], the squared wavelet coherence (SWC) and wavelet coherence phase difference (WCPD) techniques are applied to study the relationship between the coronavirus panic index and the moves of the commodity prices. Their results suggest that high correlation between a systemic event such as the Covid-19 pandemic and the commodity market volatility has been observed.

Our study differentiates from existing studies from two aspects. Firstly, extant research focuses on the risk spillover between commodity markets (see Ji et al., [24]; Shahzad et al., [25]; Mensi et al., [26]). Those doctrines mainly adopt volatility as the risk measure to explore the spillover effect between commodity markets. In complementary, our research not only uses volatility but also the market illiquidity risk in our model to broaden the investigation of risk spillover among financial markets. More importantly, we employ the VAR (Vector Auto regression) model to analyze the commodity market sensitivity towards the shocks of stock market risk and illiquidity as the commodity market responsiveness. As in Andrikopoulos et al. [2]'s study, their focus is on stock markets risk spillover in different countries. They unveil Granger causal relationship between risk, return and illiquidity across G7 stock market and they deliver the research implications regarding the international portfolio diversification. In contrast, our focal point rests on the interaction between commodity market and stock market and also, our data sample concentrates in Chinese markets instead of G7 stock markets.

Through our study, we employ the VAR model to analyze the impact persistence of volatility spillover and illiquidity spillover, especially from the stock market towards the commodity market. First, a cross-market spillover of volatility and illiquidity in Chinese financial markets is exhibited in this paper. We use the Granger test to demonstrate a noticeable interaction between volatility and illiquidity in different financial markets. More interestingly, we find that different commodity markets have different responsiveness to stock market shocks. We thereby categorize commodity markets into three groups according to their shock responsiveness behavior. The volatility and illiquidity of most commodity markets are self-dependent, which we denote as the first group, and the other two groups are prone to respond to stock market shocks.

In particular, we unveil that copper market volatility responds strongly to stock market illiquidity shocks and that the shock impact is persistent. Nevertheless, the illiquidity of the copper market is unresponsive to stock market volatility shocks. On the other hand, stock market volatility shocks induce illiquidity behavior in the corn and soybean markets, which can be recognized as agricultural markets, and the stock market shocks on agricultural markets are persistent. Consequently, we classify the copper market and the agricultural market as two groups of markets excluding self-dependent markets, and thus, we sustain that commodity markets hold different characteristics. Our results indicate that agricultural markets are heavily regulated by the government, suggesting that trading activities may be largely affected by stock market policy movement (Ding and Zhang [27]) since investors are concerned about policy uncertainty (Sinha [28]). On the other hand, the copper market is a distinct market that is

mainly for investment and hedging purposes (Barbi and Romagnoli [29]), and hedgers pay close attention to market liquidity since sufficient market liquidity is a vital requisite for successful hedging. Thus, we propose that illiquidity in the stock market may have a considerable effect on copper market volatility. Market sensitivity enables policy makers to formulate corresponding policies according to different degrees of market policy sensitivity.

The paper is organized as follows. In section 2, we describe the sample data and variable measures for commodity markets. In section 3, we present our baseline results and lay the foundation for further data analysis. Section 4 presents our empirical results and robustness check, revealing the impact persistence of volatility spillover and illiquidity spillover from the stock market to commodity markets. Section 5 concludes the paper.

## 2 Data and methodology

### 2.1 Data and variable estimations

The sample data that we collect are for the Chinese commodity markets and the Chinese stock market on a weekly basis from the WIND database. The sample covers the period from January 1, 2010 to March 22, 2021 (575 total weekly observations for each series), which covers the most recent 11 years of data in the Chinese financial markets.

For the variable description, the superscript 'coa' represents coal commodity futures, 'cop' represents copper commodity futures, 'cor' represents corn commodity futures, 'gol' represents gold commodity futures, 'met' represents methanol commodity futures, 'oil' represents oil commodity futures, 'soy' represents soybean commodity futures, 'ste' represents steel commodity futures, 'sug' represents sugar commodity futures and 'csi' represents the CSI 300 Index for the Chinese stock market. The CSI 300 Index consists of the 300 largest and most liquid A-share stocks and aims to reflect the overall performance of the Chinese A-share market. Therefore, we select this index to represent the movement of the Chinese stock market. For the three main variable estimations, $L$ stands for the Amihud measure of illiquidity, $V$ stands for return volatility of commodity futures and $r$ stands for the realized return of commodity futures.

For the empirical analysis, we use commodity prices ($P_t^i$) from the Chinese commodity futures markets to find the realized returns of different commodity futures markets, denoted as $r_t^i$, which is defined as $r_t^i = \left( \frac{P_t^i - P_{t-1}^i}{P_{t-1}^i} \right)$. Then, we use the past one-month standard deviation of the return to proxy for the market volatility estimations.

We finally define the market illiquidity proxy ($L_t$), which measures the illiquidity of the market. This indicates that the smaller the Lt is, the more liquid the market is. Marshall et al. [30] test a large number of liquidity proxies for 19 commodities. They find that the Amihud liquidity proxy has the maximal correction ratio among all proxies and they strongly recommend researchers to use this proxy when modeling commodity liquidity. Kang and Zhang [31] reveal that the adjusted Amihud measure provides helpful insights for emerging financial market liquidity. Diaz and Escribano [32] show that Amihud measure is a three dimensions of liquidity measure, including depth, immediacy and tightness. As a result, we use the proxy mentioned in Amihud [33] to measure asset illiquidity for this study, which takes the following form:

$$L_t = \frac{|R_t|}{Vol_t} \tag{1}$$

where $R_t$ is the asset return at time $t$ and $Vol_t$ is the asset trading volume at time $t$.

As a result, realized weekly returns, monthly volatility and weekly illiquidity measures are the three major variables for the empirical analysis in this paper.

## 2.2 VAR model

The main methodology we adopt in this paper is the vector autoregression (VAR) model, which is an ad hoc dynamic multi-variable model analyzing a number of endogenous variables simultaneously, where each endogenous variable is regressed on its own lags and the lags of all other variables (Sims [34]; George et al., [35]; Marcondes and Valk [36]). The model is also widely applied in studying the relations among different commodity markets (see Alsalman [37]; Dolatabadi et al., [38]; Zhang and Lin [39]). The VAR model is extremely useful for jointly describing the dynamical behavior of financial time series and can express the evolution of a set of K endogenous variables as a linear function of their past values. The basic VAR(p) model takes the following form (for $X_t$ is the vector of endogenous variables concerned):

$$\mathbf{X_t} = \theta_0 + \sum_{j=1}^{p} \theta_j \mathbf{X_{t-j}} + \varepsilon_t, \tag{2}$$

where $\theta_0$ is a K*1 vector of constants, $\theta_j$ for j = 1, . . ., p, is a K*K matrix of model coefficients and $\varepsilon_t$ is a K*1 vector of IID (Independent and Identically Distributed) Gaussian residuals terms for the VAR model.

## 3 Baseline empirical results

This section presents our empirical results on the impact persistence of illiquidity spillover and volatility spillover among different commodity markets. Our results are sequentially assigned to a series of tables.

Table 1 presents the statistical properties of our dataset in terms of market illiquidity and volatility, which are averaged across stocks for each financial market. The descriptive statistics presented are the mean, maximum, minimum, and standard deviation of the time series of commodity market illiquidity and volatility over the sample period. Panel A in Table 1 presents the statistical summary for illiquidity, Panel B presents the statistical summary for volatility, and we collect a total of 575 observations for each financial market. From those observations, it can be seen that the steel market has the highest illiquidity, and oil has the highest volatility (an Amihud measure of 3.3% and 3.1% volatility in terms of weekly estimation). Moreover, the well-known features of asymmetry and fat tails in illiquidity series are also present in the volatility series for the nine commodity markets.

Before we construct our VAR model, it is essential to ensure the stability of the illiquidity and volatility time series. The unit root test can provide solid evidence that our time series are stationary. From Table 2, we can find that all time series are stationary, and thus, there is no spurious regression problem in the illiquidity and volatility time series. Therefore, our dataset is suitable for further methodological analysis, such as correlation analysis, Granger casualty test, variance decomposition analysis and impulse response analysis.

Since our study is designated for spillover analysis, a close correlation between two markets is one of the key prerequisites for illiquidity and volatility spillover among different markets. Table 3.1 shows evidence of a significantly positive correlation of the illiquidity measures across some commodity markets, especially between the markets of steel and copper, steel and gold, and steel and soybean (45.6%, 39.9% and 34.6%, respectively). This reflects the intuition that similar commodities, like metals, may have positive correlations regarding the market illiquidity. Table 3.2 shows evidence of a significantly positive correlation of the volatility

**Table 1. Descriptive statistics of illiquidity and volatility.** This table presents the mean, standard deviation (SD) for the illiquidity and volatility of each commodity market as well as the stock market. Illiquidity is measured using the Amihud measure for each market. The sample runs from January 1, 2010 to March 22, 2021.

| | Mean | Max | Min | Std. Dev. | Obs |
|---|---|---|---|---|---|
| Panel A: Illiquidity | | | | | |
| $L_t^{coa}$ | 0.011180 | 1.000000 | 0.000000 | 0.076063 | 575 |
| $L_t^{cop}$ | 0.002000 | 0.035770 | 0.000000 | 0.002931 | 575 |
| $L_t^{cor}$ | 0.031932 | 0.404272 | 0.000000 | 0.042154 | 575 |
| $L_t^{gol}$ | 0.014279 | 0.180103 | 0.000000 | 0.022790 | 575 |
| $L_t^{met}$ | 0.024640 | 1.000000 | 0.000000 | 0.084928 | 575 |
| $L_t^{oil}$ | 0.021959 | 1.000000 | 0.000000 | 0.081798 | 575 |
| $L_t^{soy}$ | 2.47E-07 | 3.01E-06 | 0.000000 | 3.23E-07 | 575 |
| $L_t^{ste}$ | 0.033425 | 1.000000 | 0.000000 | 0.055709 | 575 |
| $L_t^{sug}$ | 0.009579 | 0.165720 | 0.000000 | 0.015365 | 575 |
| $L_t^{csi}$ | 0.295441 | 1.000000 | 0.086581 | 0.032797 | 575 |
| Panel B: Volatility | | | | | |
| $V_t^{coa}$ | 0.014775 | 0.083786 | 0.000000 | 0.015271 | 575 |
| $V_t^{cop}$ | 0.022256 | 0.106783 | 0.000928 | 0.014866 | 575 |
| $V_t^{cor}$ | 0.013463 | 0.096530 | 0.000803 | 0.012528 | 575 |
| $V_t^{gol}$ | 0.017606 | 0.070106 | 0.002538 | 0.010609 | 575 |
| $V_t^{met}$ | 0.022170 | 0.080071 | 0.000000 | 0.016013 | 575 |
| $V_t^{oil}$ | 0.030537 | 0.155360 | 0.000000 | 0.027962 | 575 |
| $V_t^{soy}$ | 0.021500 | 0.077688 | 0.002133 | 0.011998 | 575 |
| $V_t^{ste}$ | 0.023534 | 0.119738 | 0.000896 | 0.016064 | 575 |
| $V_t^{sug}$ | 0.018758 | 0.068504 | 0.002073 | 0.009989 | 575 |
| $V_t^{csi}$ | 0.419591 | 3.690484 | 0.031420 | 0.581031 | 575 |

measures across most commodity markets in China, which indicates that volatility can be transmitted from one market to another and that those markets usually have similar moving trends. However, some negative relationships are also observed for volatility series, like oil market and CSI index, gold market with coal and steel markets. Gold price movement might be perceived as negatively correlated with the prosperity of the real estate market; thus, the gold market may be negatively correlated with the coal and steel markets, which are closely related to the real estate industry.

As most of our time series have close relations with each other, we undertake a stronger analysis, which is the Granger causality test. The Granger test was developed by Granger [40] to study the causality between two time series. This method is frequently used in scrutinizing casual relations between commodity markets, such as the relationship between oil prices and other agricultural commodity prices (Nazlioglu and Soytas [41]). The Granger causality relation is a stronger relation than our aforementioned correlation test. More importantly, the Granger causality test is a prerequisite for VAR analysis (see Massa and Rosellon [42]).

The Granger causality results are presented in Table 4.1 and 4.2. In particular, we report the results of individual VAR estimations for each market over the whole sample. We test the null hypothesis that variable i does not Granger cause variable j, i.e., that the lag coefficients of variable i are jointly zero when variable j is the dependent variable. Specifically, the Granger causality is from market j to market i. Table 4.1 and 4.2 report the p-values of the above test hypotheses, where i is the row variable and j is the column variable. Table 4.1 and 4.2 report the p-values of the above test hypotheses, where i is the column variable and j is the row

**Table 2. Unit root test of illiquidity and volatility.** The table presents the individual unit root test results for the illiquidity and volatility of each market. Illiquidity is measured using the Amihud measure for each financial market. The sample runs from January 1, 2010 to March 22, 2021.

| Series | Prob. | Lag | Max Lag | Obs |
|--------|-------|-----|---------|-----|
| Panel A: Illiquidity | | | | |
| $L_t^{coa}$ | 0.0001 | 16 | 18 | 558 |
| $L_t^{cop}$ | 0.0000 | 2 | 18 | 569 |
| $L_t^{cor}$ | 0.0004 | 2 | 18 | 562 |
| $L_t^{gol}$ | 0.0338 | 18 | 18 | 556 |
| $L_t^{met}$ | 0.0080 | 7 | 18 | 567 |
| $L_t^{oil}$ | 0.0180 | 17 | 18 | 557 |
| $L_t^{soy}$ | 0.0000 | 3 | 18 | 571 |
| $L_t^{ste}$ | 0.0000 | 2 | 18 | 572 |
| $L_t^{sug}$ | 0.0000 | 0 | 18 | 574 |
| $L_t^{csi}$ | 0.0000 | 0 | 18 | 574 |
| Panel B: Volatility | | | | |
| $V_t^{coa}$ | 0.0002 | 17 | 18 | 557 |
| $V_t^{cop}$ | 0.0000 | 8 | 18 | 566 |
| $V_t^{cor}$ | 0.0000 | 12 | 18 | 562 |
| $V_t^{gol}$ | 0.0000 | 4 | 18 | 570 |
| $V_t^{met}$ | 0.0000 | 15 | 18 | 559 |
| $V_t^{oil}$ | 0.0000 | 8 | 18 | 566 |
| $V_t^{soy}$ | 0.0001 | 4 | 18 | 570 |
| $V_t^{ste}$ | 0.0000 | 4 | 18 | 570 |
| $V_t^{sug}$ | 0.0004 | 4 | 18 | 570 |
| $V_t^{csi}$ | 0.0000 | 4 | 18 | 560 |

variable. It can be observed that all time series exhibit at least a one-way Granger causality relation with another market. As a result, the VAR model can be constructed on this basis. We document a (Granger) causal effect of illiquidity on volatility in the soybean market and sugar markets, which corroborates the hypothesis that the principal source of volatility is the transmission of information through trading activity (and trading activity is a fundamental determinant of illiquidity). The significant (causal) effect of volatility on illiquidity—largely associated with the risks that market makers undertake—is present in the corn and sugar markets.

Therefore, we can argue that information transmission between volatility and illiquidity exists in a large number of commodity markets. These empirical results also indicate that the structural interdependence of illiquidity and volatility is extensively present across almost all commodity markets: illiquidity in the coal market Granger causes volatility and illiquidity in the oil market. Illiquidity in the copper market Granger causes volatility in the oil market. Regarding the stock market, volatility of CSI 300 Index is the Granger cause for illiquidity in the copper, gold, steel and sugar markets, as well as the cause for volatility in the methanol market. The illiquidity of soybean market, for another example, is the Granger cause for the volatility of coal, copper, methanol and soybean markets. It is also the Granger cause for the gold market illiquidity. Such results are largely expected since major commodity markets are substantially integrated, as shown in our results and a plethora of existing studies (see Morana [43]; Jacks et al., [44]; Chiou-Wei et al., [45]). These results support further VAR analysis, which we present in section 4.

**Table 3. The correlation matrix.** The table presents the correlation among illiquidity series and volatility series for all financial markets. The sample runs from January 1, 2010 to March 22, 2021.

(a) Table 3.1: Correlation analysis of the illiquidity measurement.

| | $L_t^{coa}$ | $L_t^{cop}$ | $L_t^{cor}$ | $L_t^{gol}$ | $L_t^{met}$ | $L_t^{oil}$ | $L_t^{soy}$ | $L_t^{ste}$ | $L_t^{sug}$ | $L_t^{csi}$ |
|---|---|---|---|---|---|---|---|---|---|---|
| $L_t^{coa}$ | 1.0000 | | | | | | | | | |
| | — | | | | | | | | | |
| $L_t^{cop}$ | 0.0016 | 1.0000 | | | | | | | | |
| | 0.9701 | — | | | | | | | | |
| $L_t^{cor}$ | 0.1806 | -0.0332 | 1.0000 | | | | | | | |
| | 0.0000 | 0.4269 | — | | | | | | | |
| $L_t^{gol}$ | -0.0415 | 0.1822 | 0.1128 | 1.0000 | | | | | | |
| | 0.3203 | 0.0000 | 0.0068 | — | | | | | | |
| $L_t^{met}$ | -0.0404 | -0.0559 | 0.1111 | -0.0127 | 1.0000 | | | | | |
| | 0.3337 | 0.1810 | 0.0076 | 0.7621 | — | | | | | |
| $L_t^{oil}$ | 0.0410 | -0.0413 | -0.0077 | -0.0480 | 0.1042 | 1.0000 | | | | |
| | 0.3265 | 0.3223 | 0.8542 | 0.2508 | 0.0125 | — | | | | |
| $L_t^{soy}$ | -0.0750 | 0.1860 | 0.0857 | 0.1543 | 0.0516 | -0.0730 | 1.0000 | | | |
| | 0.0723 | 0.0000 | 0.0399 | 0.0002 | 0.2165 | 0.0801 | — | | | |
| $L_t^{ste}$ | -0.0076 | 0.4565 | 0.0593 | 0.3992 | -0.0332 | -0.0531 | 0.3463 | 1.0000 | | |
| | 0.8566 | 0.0000 | 0.1555 | 0.0000 | 0.4262 | 0.2035 | 0.0000 | — | | |
| $L_t^{sug}$ | -0.0036 | 0.2923 | 0.0097 | -0.0252 | 0.1056 | -0.0379 | 0.1214 | 0.1200 | 1.0000 | |
| | 0.9323 | 0.0000 | 0.8161 | 0.5458 | 0.0113 | 0.3647 | 0.0036 | 0.0040 | — | |
| $L_t^{csi}$ | -0.0091 | 0.0040 | 0.0014 | -0.0127 | 0.0257 | -0.0082 | 0.0137 | -0.0087 | 0.0723 | 1.0000 |
| | 0.8272 | 0.9231 | 0.9727 | 0.7617 | 0.5386 | 0.8449 | 0.7423 | 0.8356 | 0.0834 | — |

(b) Table 3.2: Correlation analysis of the volatility measurement.

| | $V_t^{coa}$ | $V_t^{cop}$ | $V_t^{cor}$ | $V_t^{gol}$ | $V_t^{met}$ | $V_t^{oil}$ | $V_t^{soy}$ | $V_t^{ste}$ | $V_t^{sug}$ | $V_t^{csi}$ |
|---|---|---|---|---|---|---|---|---|---|---|
| $V_t^{coa}$ | 1.0000 | | | | | | | | | |
| | — | | | | | | | | | |
| $V_t^{cop}$ | -0.0706 | 1.0000 | | | | | | | | |
| | 0.0909 | — | | | | | | | | |
| $V_t^{cor}$ | 0.1811 | 0.1553 | 1.0000 | | | | | | | |
| | 0.0000 | 0.0002 | — | | | | | | | |
| $V_t^{gol}$ | -0.2693 | 0.4377 | 0.0350 | 1.0000 | | | | | | |
| | 0.0000 | 0.0000 | 0.4024 | — | | | | | | |
| $V_t^{met}$ | 0.5146 | -0.1333 | 0.1693 | -0.0924 | 1.0000 | | | | | |
| | 0.0000 | 0.0014 | 0.0000 | 0.0268 | — | | | | | |
| $V_t^{oil}$ | 0.1958 | 0.1012 | 0.1630 | 0.2107 | 0.3477 | 1.0000 | | | | |
| | 0.0000 | 0.0152 | 0.0001 | 0.0000 | 0.0000 | — | | | | |
| $V_t^{soy}$ | 0.1472 | 0.0773 | 0.1024 | -0.0290 | 0.1110 | 0.0689 | 1.0000 | | | |
| | 0.0004 | 0.0640 | 0.0140 | 0.4871 | 0.0077 | 0.0987 | — | | | |
| $V_t^{ste}$ | 0.3598 | 0.2253 | 0.2148 | -0.0821 | 0.2430 | 0.1091 | 0.2930 | 1.0000 | | |
| | 0.0000 | 0.0000 | 0.0000 | 0.0490 | 0.0000 | 0.0088 | 0.0000 | — | | |
| $V_t^{sug}$ | -0.1087 | 0.2495 | 0.0542 | 0.1394 | -0.0484 | 0.0432 | 0.0339 | -0.0192 | 1.0000 | |
| | 0.0091 | 0.0000 | 0.1941 | 0.0008 | 0.2464 | 0.3014 | 0.4176 | 0.6466 | — | |
| $V_t^{csi}$ | -0.0442 | -0.0679 | -0.0617 | 0.0088 | -0.0384 | -0.0420 | 0.0244 | -0.0706 | -0.0709 | 1.0000 |
| | 0.2897 | 0.1040 | 0.1398 | 0.8338 | 0.3576 | 0.3152 | 0.5585 | 0.0908 | 0.0894 | — |

**Table 4. VAR estimation results for pairs of markets: Granger causality tests.** Those two tables present results from a six-variable VAR for the cross-section of pair of markets i and j. The horizontal row represents market i, which is the dependent variable and the vertical row represents market j, which is the independent variable. The Granger causality is from market j to market i. p-Values of the null hypothesis that the column variable does not Granger cause the row variable are presented for selected markets. Illiquidity is measured using the Amihud measure for each market. All variables are adjusted for deterministic time series variations. We choose the number of lags based on the SC and HQ criteria. The sample runs from January 1, 2010 to March 22, 2021. ** denotes significance at the 5% level and *denotes significance at the 10% level.

| (a) Table 4.1: VAR estimation results for pairs of markets: Granger causality tests part 1. | | | | | | | | | | |
|---|---|---|---|---|---|---|---|---|---|---|
| Market | | COA | | COP | | COR | | GOL | | MET | |
| | | Illiquidity | Volatility | Illiquidity | Volatility | Illiquidity | Volatility | Illiquidity | Volatility | Illiquidity | Volatility |
| COA | Illiquidity | | 0.9323 | 0.5952 | 0.9538 | 0.0008** | 0.0003** | 0.9391 | 0.9936 | 0.5188 | 0.5604 |
| | Volatility | 0.4234 | | 0.4968 | 0.5912 | 0.0114* | 0.0109* | 6.00E-08** | 0.0020** | 0.0460* | 0.0012** |
| COP | Illiquidity | 0.4752 | 0.4614 | | 0.0616 | 0.2805 | 0.2845 | 0.5278 | 0.0109* | 0.1531 | 0.8149 |
| | Volatility | 0.9626 | 0.3901 | 0.3627 | | 0.0900 | 0.8911 | 0.323 | 0.1734 | 0.9542 | 0.9475 |
| COR | Illiquidity | 0.1160 | 0.4635 | 0.6471 | 0.0436* | | 0.0157* | 0.7059 | 0.2537 | 0.7388 | 0.3915 |
| | Volatility | 0.8065 | 0.0519 | 0.3750 | 0.0608 | 1.00E-05** | | 0.1432 | 0.3922 | 0.1318 | 0.0043** |
| GOL | Illiquidity | 0.6360 | 0.0157 * | 0.7160 | 0.0914 | 0.1305 | 0.7250 | | 0.0312* | 0.4257 | 0.0008** |
| | Volatility | 0.7605 | 0.1667 | 0.5763 | 0.0801 | 0.2923 | 0.7289 | 0.1526 | | 0.0059** | 0.4422 |
| MET | Illiquidity | 0.6381 | 0.5717 | 0.0693 | 0.6725 | 0.2661 | 0.3553 | 0.7404 | 0.835 | | 0.7215 |
| | Volatility | 0.4527 | 0.0005** | 0.8907 | 0.0760 | 0.0024** | 0.586 | 2.00E-11** | 0.0644 | 0.0934 | |
| OIL | Illiquidity | 0.0389 * | 0.3568 | 0.5816 | 0.7338 | 0.0515 | 0.946 | 0.3102 | 0.9031 | 0.5574 | 0.1352 |
| | Volatility | 0.0072 ** | 0.3728 | 0.0002** | 0.7078 | 0.2315 | 0.9398 | 0.0228* | 0.2065 | 0.4027 | 0.0020** |
| SOY | Illiquidity | 0.7913 | 0.8057 | 0.2279 | 0.0670 | 0.3632 | 0.4163 | 0.2045 | 0.3548 | 0.7462 | 0.7873 |
| | Volatility | 0.7390 | 0.3432 | 0.5823 | 0.8836 | 0.7981 | 0.9998 | 0.074 | 0.1963 | 0.6719 | 0.5119 |
| STE | Illiquidity | 0.5264 | 0.9482 | 0.1757 | 0.1366 | 0.5809 | 0.3513 | 0.4771 | 0.3078 | 0.5012 | 0.9795 |
| | Volatility | 0.1859 | 0.0538 | 0.5443 | 0.6037 | 0.0121* | 0.7074 | 0.0877 | 0.2022 | 0.0168* | 0.1574 |
| SUG | Illiquidity | 0.9986 | 0.0012** | 0.0340 * | 0.2223 | 0.5603 | 0.2192 | 0.0457* | 0.1121 | 0.7556 | 0.1308 |
| | Volatility | 0.1670 | 0.9194 | 0.6428 | 0.126 | 0.0022** | 0.7553 | 0.0696 | 0.4513 | 0.0643 | 0.2889 |
| CSI | Illiquidity | 0.9426 | 0.5299 | 0.7272 | 0.7639 | 0.9377 | 0.6958 | 0.1974 | 0.5274 | 0.7528 | 0.0743 |
| | Volatility | 0.4931 | 0.7391 | 0.1799 | 0.6521 | 0.7482 | 0.5925 | 0.6438 | 0.6301 | 0.3731 | 0.6121 |

| (b) Table 4.2: VAR estimation results for pairs of markets: Granger causality tests part 2. | | | | | | | | | | |
|---|---|---|---|---|---|---|---|---|---|---|
| Market | | OIL | | SOY | | STE | | SUG | | CSI | |
| | | Illiquidity | Volatility | Illiquidity | Volatility | Illiquidity | Volatility | Illiquidity | Volatility | Illiquidity | Volatility |
| COA | Illiquidity | 0.5293 | 0.1997 | 0.4361 | 0.5381 | 0.7479 | 0.9186 | 0.1649 | 0.0418* | 0.8577 | 0.7293 |
| | Volatility | 0.2415 | 0.0666 | 0.0090** | 0.7383 | 0.0646 | 0.0019** | 0.6887 | 0.1021 | 0.6395 | 0.2880 |
| COP | Illiquidity | 0.3327 | 0.1368 | 0.9791 | 0.8725 | 0.2186 | 0.0845 | 0.4093 | 0.3649 | 0.8948 | 1.00E-06** |
| | Volatility | 0.9324 | 0.4095 | 0.0038** | 0.4581 | 0.4628 | 0.2160 | 0.2754 | 0.5249 | 0.8045 | 0.5160 |
| COR | Illiquidity | 0.2229 | 0.0144* | 0.8724 | 0.7905 | 0.2476 | 0.0834 | 0.8510 | 0.1340 | 0.7680 | 0.4773 |
| | Volatility | 4.00E-08** | 0.0074** | 0.4281 | 0.2642 | 0.6896 | 0.0358* | 0.6034 | 0.1465 | 0.4966 | 0.2024 |
| GOL | Illiquidity | 0.2321 | 0.3130 | 0.0027** | 0.5907 | 0.0010** | 0.3482 | 0.1554 | 0.0211* | 0.4109 | 0.0019** |
| | Volatility | 0.4247 | 0.7575 | 0.4207 | 0.7112 | 0.3814 | 0.6900 | 0.0093** | 0.8280 | 0.6193 | 0.4778 |
| MET | Illiquidity | 0.0364* | 0.8428 | 0.4889 | 0.5605 | 0.2980 | 0.2974 | 0.8640 | 0.8646 | 0.9559 | 0.1436 |
| | Volatility | 0.0449* | 0.0947 | 0.0010** | 0.8805 | 0.0172* | 0.3675 | 0.5294 | 0.1036 | 0.3125 | 0.0393* |
| OIL | Illiquidity | | 0.0441* | 0.4153 | 0.6136 | 0.8989 | 0.9085 | 0.6226 | 0.2701 | 0.7966 | 0.9395 |
| | Volatility | 0.0319* | | 0.2215 | 0.5312 | 0.9760 | 0.5045 | 0.0232* | 0.8143 | 0.4391 | 0.1814 |
| SOY | Illiquidity | 0.2486 | 0.5331 | | 0.1822 | 0.6135 | 0.8526 | 0.1141 | 0.2942 | 0.9279 | 2.00E-05** |
| | Volatility | 0.4677 | 0.9827 | 0.0487* | | 0.7129 | 0.0131* | 0.2918 | 0.2377 | 0.3314 | 0.3316 |
| STE | Illiquidity | 0.3943 | 0.3430 | 0.9719 | 0.7822 | | 0.0375* | 0.8695 | 0.8224 | 0.8131 | 4.00E-06** |
| | Volatility | 0.1623 | 0.2907 | 0.0411* | 0.6616 | 0.0060** | | 0.8420 | 0.8557 | 0.9084 | 0.6717 |
| SUG | Illiquidity | 0.9044 | 0.1186 | 0.4360 | 0.7430 | 0.6187 | 0.7135 | | 0.0327* | 0.6250 | 8.00E-10** |
| | Volatility | 0.4748 | 0.3324 | 0.4114 | 0.5621 | 0.9382 | 0.1529 | 0.0040** | | 0.6778 | 0.2221 |
| CSI | Illiquidity | 0.2861 | 0.5466 | 0.7489 | 0.8478 | 0.5036 | 0.5038 | 0.5596 | | 0.4981 | 0.7391 |
| | Volatility | 0.7962 | 0.8376 | 0.5117 | 0.6676 | 0.6437 | 0.2373 | 0.9410 | 2.00E-06 | | 0.9929 |

## 4 Further empirical results

### 4.1 Further VAR analysis

In section 3, we evaluated our dataset's potential for further analysis. We used a unit root test to verify the stationarity of our time series, and we used a correlation matrix to confirm the close relationship among different markets. We also adopted the Granger test to ensure that at least one-way Granger causality exists between two markets. On this basis, we provide further analysis in this section to examine the impact persistence of volatility spillover and illiquidity spillover in commodity markets by utilizing the VAR model. The VAR model includes all commodity markets, which excludes the possible commodity market shocks from stock market shocks; thus, we now investigate the impact of stock market shocks.

To use the VAR model, it is essential to determine the optimal lag. Table 5 presents our VAR model optimal lag selection. We mix both illiquidity and volatility into one VAR model, and we use five tests for optimal lag selection. It can be observed that one time lag has two optimal indicators, the Schwarz information criterion and Hannan-Quinn information criterion. Other time lags, such as the seventh and eighth lags, only have one optimal indicator, except the fifth lag. We then run both one time lag and fifth time lag where the one lag regression delivers better results in terms of model fitness. As a result, we use one time lag to construct our VAR model. After VAR model construction, we utilize variance decomposition and the impulse function to demonstrate the impact persistence of volatility spillover and illiquidity spillover in commodity markets.

Another method to shed more light on the dynamics of illiquidity and volatility spillovers is to assess corresponding variance decompositions. Variance decomposition splits variance into slices and unveils the most important impact factor over different time horizons beyond the selected time period. The key feature of variance decomposition is that it is insensitive to the variable ordering since the ordering is determined by our VAR system (Pesaran and Shin [46]). Variance decomposition can also lead to a deep analysis of the VAR model result, showing the impact of each innovation series (Engle and Granger [47]). Based on this method, we take advantage of our VAR model to further explore the causal relationship between volatility and illiquidity in the commodity market. We decompose the variance for commodity market volatility and illiquidity, and the results are presented in Table 6.

**Table 5. VAR optimal lag selection.** This table presents our VAR model optimal lag selection criteria. We mixed both illiquidity and volatility into one VAR model and we use five tests for optimal lag selection. LR is the sequential modified LR test statistic (each test at 5% level); FPE is the Final prediction error; AIC is the Akaike information criterion; SC is Schwarz information criterion; HQ is Hannan-Quinn information criterion. * indicates the optimal section based on the corresponding lag selection criteria. The sample runs from January 1, 2010 to March 22, 2021.

| VAR Lag Order Selection | | | | | | |
|---|---|---|---|---|---|---|
| Sample: 01/01/2010-03/22/2021 | | | | | | |
| Included Obs: 567 | | | | | | |
| Lag | LogL | LR | FPE | AIC | SC | HQ |
| 0 | 32669.84 | NA | 2.16e-75 | -115.167 | -115.0139 | -115.1072 |
| 1 | 36067.49 | 6543.612 | 5.52e-80 | -125.7407 | -122.5256* | -124.4860* |
| 2 | 36471.04 | 748.7380 | 5.48e-80 | -125.7532 | -119.4761 | -123.3035 |
| 3 | 36828.69 | 638.3490 | 6.44e-80 | -125.6038 | -116.2648 | -121.9592 |
| 4 | 37316.72 | 836.6301 | 4.83e-80 | -125.9144 | -113.5133 | -121.0747 |
| 5 | 37851.27 | 878.6506 | 3.12e-80* | -126.3889* | -110.9259 | -120.3543 |
| 6 | 38192.74 | 537.2025* | 4.06e-80 | -126.1825 | -107.6575 | -118.9529 |
| 7 | 38487.53 | 442.9629 | 6.37e-80 | -125.8114 | -104.2244 | -117.3868 |
| 8 | 38753.04 | 380.2388 | 1.14e-79 | -125.337 | -100.688 | -115.7175 |

In panel A, we can see that shocks in a market's own market illiquidity are the most important factor by far, explaining, on average, 100.00% of the variance over short horizons. This fraction declines by maximally 11.96% for longer forecasting horizons. The fraction of the error variance in forecasting illiquidity due to shocks in CSI 300 illiquidity is negligible for short forecast horizons and increases from 0 to maximally 1.23% for longer forecast horizons, while the volatility increases from 0 to maximally 2.81%. These results suggest that shocks in own market illiquidity are the most important factor in explaining illiquidity dynamics. Own market volatility is the next most important variable for both short-term and long-term forecast horizons. Little illiquidity spillover can be found from the stock market. The largest illiquidity spillover from the stock market is 1.23% to the gold market, followed by 0.65% to the steel market. Compared with illiquidity spillover, stock market volatility has a stronger effect on commodity market illiquidity, such as in the coal market and in the corn market. It might be arguable that the fluctuation of the stock market may influence commodity illiquidity since investors may reduce their trading activities in high fluctuation periods in these two markets.

On the other hand, in panel B, we can see that volatility in the commodity market is more dependent on its own market illiquidity, even after 10 periods. There is little volatility spillover from the stock market. The copper market probably has the largest volatility spillover from the stock market, which is approximately 9.23%. Compared with the illiquidity spillover, commodity market volatility relies heavily on its own market illiquidity, especially in the corn market and soybean markets. In these two markets, as much as over 20% of the market volatility can be explained by market illiquidity after 10 periods. The maximal percentage is 34.96% for the corn market, followed by 19.69% of the soybean market. As a result, for those markets, we can use illiquidity as a trading signal. When there is a shock in market illiquidity, it is likely to have a stock in commodity returns.

Finally, we use the impulse functions and spillover index to analyze the stock market shock persistence on commodity market volatility and illiquidity. Fig 2 delivers the result of the impulse response of commodity market illiquidity to stock market illiquidity shocks. For most commodity markets, the impact of stock market illiquidity shocks is persistent. For the copper, soybean and sugar markets, the stock market shock is absorbed over 10 periods, while in other markets, the shock is persistent over 10 periods.

Fig 3 delivers the result of the impulse response of commodity market volatility to stock market volatility shock, as well as the spillover index to measure the overall market spillover. The nine commodity markets can be categorized into three groups. The first commodity group, including the copper, gold and soybean markets, has little response to the stock market shock over 10 periods. The second commodity group, including the coal, methanol, oil and sugar markets, rigorously responds to the stock market shock. The third commodity group, including the coal and steel markets, has little response to the stock market shock at the beginning, and then the response gradually strengthens over time.

Fig 4 displays the result of the impulse response of commodity market illiquidity to stock market volatility shocks. The responsiveness of different commodity markets is quite heterogeneous. Illiquidity in the gold market hardly responds to the stock market volatility shock. On the other hand, the coal, corn, oil and soybean markets exhibit considerable responsiveness to the stock market volatility shock.

Fig 5 displays the result of the impulse response of commodity market volatility to stock market illiquidity shocks. Most commodity markets are substantially responsive to the stock market illiquidity shock, especially the gold market. For the corn and soybean markets, the stock market shock is weak and is absorbed over time.

For further risk spillover analysis, we also employ the DY spillover index to investigate the volatility and illiquidity spillover among Chinese commodity markets (see Diebold and Yilmaz

**Table 6. Variance decomposition of market i illiquidity to own and other market illiquidity and volatility.** This table presents the variance decompositions computed from a six-variable VAR for market i and CSI300 market. All variables are adjusted for deterministic time series variations. Illiquidity is measured using the Amihud measure for each stock and is averaged across stocks for each market. We choose the number of lags based on the SC and HQ criteria. The sample runs from January 1, 2010 to March 22, 2021.

| Market | Period | Illiquidity of market i | Volatility of market i | CSI300 illiquidity | CSI300 volatility |
|---|---|---|---|---|---|
| Panel A: Illiquidity | | | | | |
| COA | 1 | 100.0000 | 0.000000 | 0.000000 | 0.000000 |
| | 10 | 99.37505 | 0.364814 | 0.014484 | 0.245655 |
| COP | 1 | 100.0000 | 0.000000 | 0.000000 | 0.000000 |
| | 10 | 95.33096 | 1.523445 | 0.328197 | 2.817398 |
| COR | 1 | 100.0000 | 0.000000 | 0.000000 | 0.000000 |
| | 10 | 88.04464 | 9.539225 | 0.191182 | 2.224954 |
| GOL | 1 | 100.0000 | 0.000000 | 0.000000 | 0.000000 |
| | 10 | 93.60720 | 3.210749 | 1.229579 | 1.952474 |
| MET | 1 | 100.0000 | 0.000000 | 0.000000 | 0.000000 |
| | 10 | 98.10826 | 1.335900 | 0.220327 | 0.335515 |
| OIL | 1 | 100.0000 | 0.000000 | 0.000000 | 0.000000 |
| | 10 | 98.62641 | 1.022318 | 0.128770 | 0.222505 |
| SOY | 1 | 100.0000 | 0.000000 | 0.000000 | 0.000000 |
| | 10 | 94.64650 | 2.589415 | 0.485255 | 2.278826 |
| STE | 1 | 100.0000 | 0.000000 | 0.000000 | 0.000000 |
| | 10 | 95.79448 | 2.733442 | 0.644992 | 0.827088 |
| SUG | 1 | 100.0000 | 0.000000 | 0.000000 | 0.000000 |
| | 10 | 94.08176 | 4.722475 | 0.292525 | 0.903240 |
| Panel B: Volatility | | | | | |
| COA | 1 | 0.005257 | 99.99474 | 0.000000 | 0.000000 |
| | 10 | 0.254523 | 91.54828 | 6.754226 | 1.442968 |
| COP | 1 | 7.893170 | 92.10683 | 0.000000 | 0.000000 |
| | 10 | 10.50027 | 76.96469 | 9.239759 | 3.295274 |
| COR | 1 | 23.09226 | 76.90774 | 0.000000 | 0.000000 |
| | 10 | 34.69026 | 64.81522 | 0.046301 | 0.448214 |
| GOL | 1 | 5.011728 | 94.98827 | 0.000000 | 0.000000 |
| | 10 | 12.86163 | 84.45334 | 1.823205 | 0.861825 |
| MET | 1 | 0.529286 | 99.47071 | 0.000000 | 0.000000 |
| | 10 | 0.472579 | 95.67415 | 1.950354 | 1.902913 |
| OIL | 1 | 0.441138 | 99.55886 | 0.000000 | 0.000000 |
| | 10 | 5.125332 | 94.28974 | 0.423431 | 0.161499 |
| SOY | 1 | 12.80407 | 87.19593 | 0.000000 | 0.000000 |
| | 10 | 19.69326 | 77.79791 | 0.644317 | 1.864513 |
| STE | 1 | 3.270837 | 96.72916 | 0.000000 | 0.000000 |
| | 10 | 9.649024 | 86.06256 | 3.158767 | 1.129645 |
| SUG | 1 | 7.691292 | 92.30871 | 0.000000 | 0.000000 |
| | 10 | 12.98080 | 83.84531 | 0.889394 | 2.284503 |

[48]; Diebold and Yilmaz [49]). Fig 6 presents the volatility spillover index using the DY spillover index method. It can be observed that the volatility spillover among different commodity markets has been relatively stable over the past 6 years, with a peak in 2020, which may be attributed to the outbreak of COVID-19. In contrast, however, the illiquidity spillover among

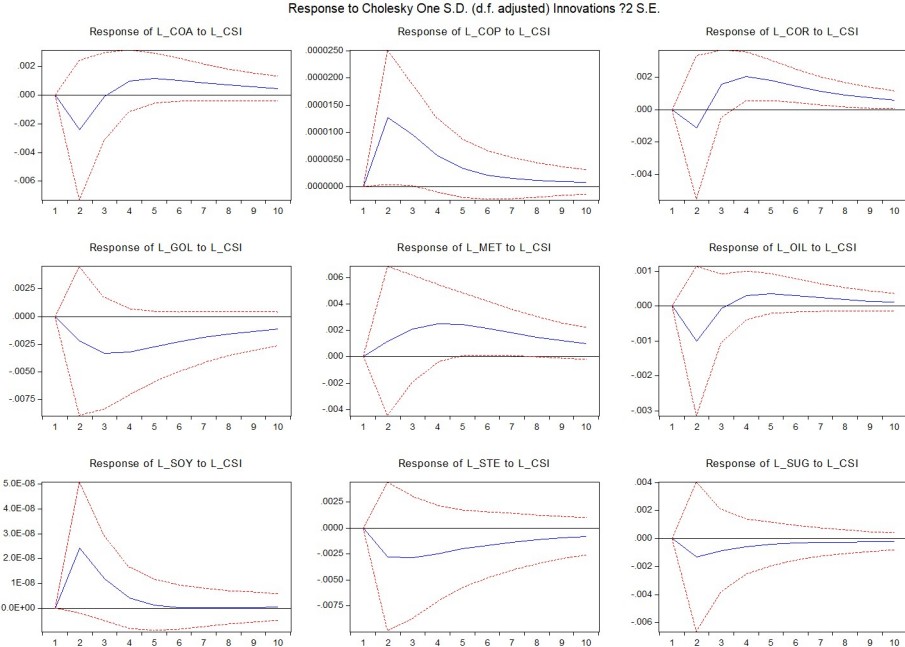

**Fig 2. Impulse response functions of market i illiquidity to CSI300 index illiquidity.** The figure presents the response of market i illiquidity (i = coa, cor, cop, gol, met, oil, soy, ste, sug) to CSI300 index illiquidity shocks. The sample runs from January 1, 2010 to March 22, 2021. The dotted lines are confidence bands based on Monte Carlo simulated standard errors.

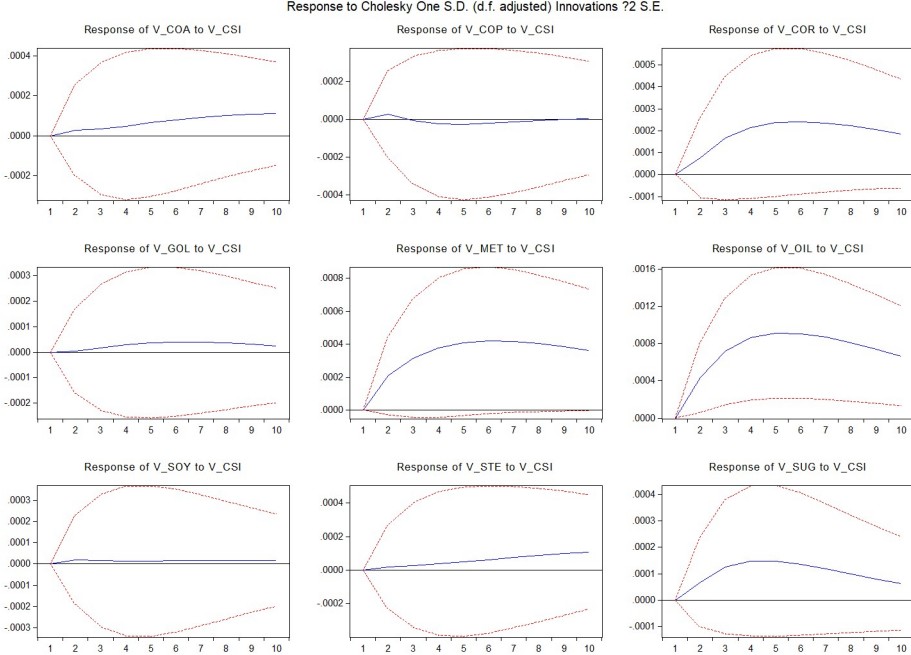

**Fig 3. Impulse response functions of market i volatility to CSI300 index volatility.** The figure presents the response of market i volatility (i = coa, cor, cop, gol, met, oil, soy, ste, sug) to CSI300 index volatility shocks. The sample runs from January 1, 2010 to March 22, 2021. The dotted lines are confidence bands based on Monte Carlo simulated standard errors.

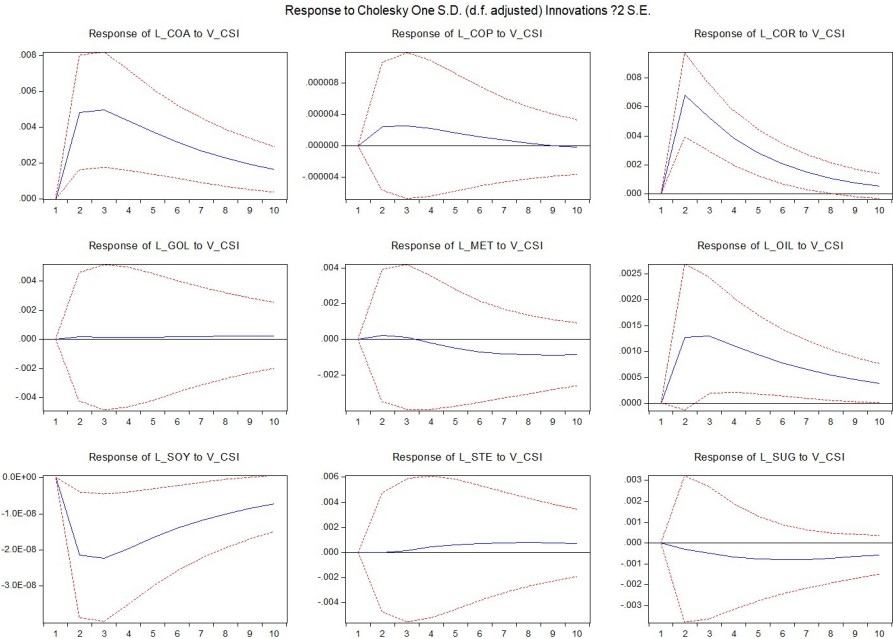

**Fig 4. Impulse response functions of market i illiquidity to CSI300 index volatility.** The figure presents the response of market i illiquidity (i = coa, cor, cop, gol, met, oil, soy, ste, sug) to CSI300 index volatility shocks. The sample runs from January 1, 2010 to March 22, 2021. The dotted lines are confidence bands based on Monte Carlo simulated standard errors.

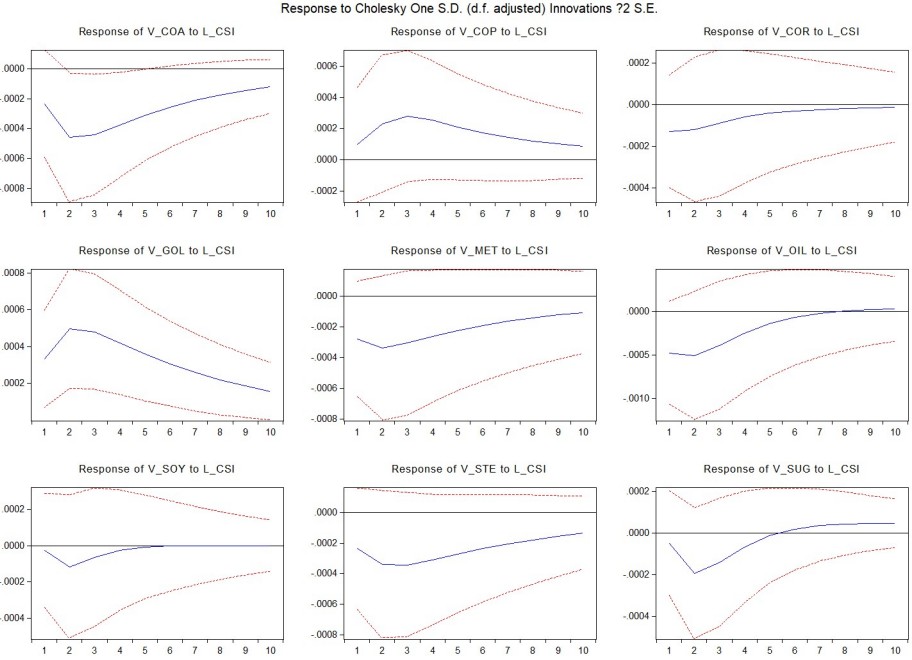

**Fig 5. Impulse response functions of market i volatility to CSI300 index illiquidity.** The figure presents the response of market i volatility (i = coa, cor, cop, gol, met, oil, soy, ste, sug) to CSI300 index illiquidity shocks. The sample runs from January 1, 2010 to March 22, 2021. The dotted lines are confidence bands based on Monte Carlo simulated standard errors.

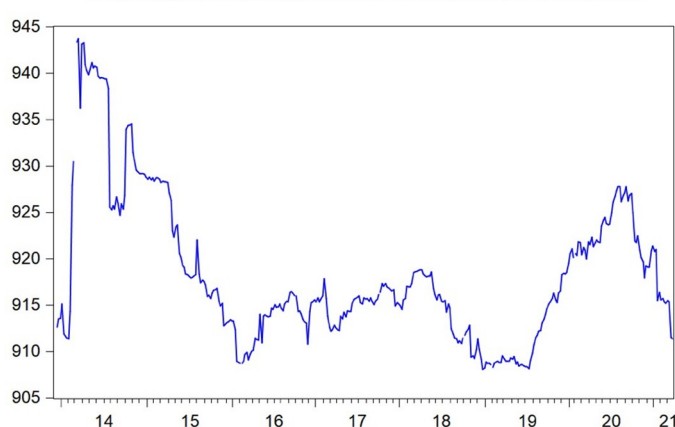

**Fig 6. Volatility spillover index plotting, estimated using the DY spillover index method for all ten financial markets.** The figure plot moving volatility spillover indexes, defined as the sum of all variance decomposition "contributions to others", estimated using a 200-week rolling window.

different commodity markets might be relatively unstable, with the peak in 2019 and early 2020 (see Fig 7). It is therefore arguable that illiquidity spillover might precede volatility spillover among different commodity markets, which can serve as a signal for policy makers to stabilize financial markets beforehand.

Further, market liquidity is a reflection of trading activities and trading fluency. Trading activities in commodity markets can be the reflection of market conditions, conveying plenty of information to policymakers. Trading activities can represent market reaction of the expected policy results in commodity futures markets. Therefore, policy impacts can be effectively spread to commodity markets from strong markets towards weak markets. More importantly, the co-integrated VAR system could be helpful in forecasting returns and volatility in different commodity markets. Since the risk spillover impact might be relatively stable, policymakers can predict the policy spillover effect from one market to another through the volatility

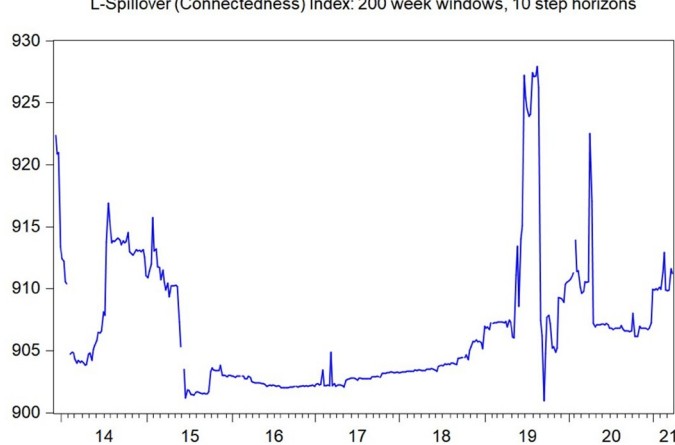

**Fig 7. Illiquidity spillover index plotting, estimated using the DY spillover index method for all ten financial markets.** The figure plot moving illiquidity spillover indexes, defined as the sum of all variance decomposition "contributions to others", estimated using a 200-week rolling window.

and illiquidity spillover channel. From the impulse response figures (Figs 2–5), it can be clear that the risk impact and its persistence for different commodity markets, which could be a crucial factor for policymakers to consider when formulating policies for the stock market.

Additionally, we further use the GARCH model to analyze the volatility clustering effect for both commodity markets and stock markets, presenting in Fig 8.

The standard GARCH (1, 1) model has the following the form:

$$\sigma_t^2 = \alpha_0 + \alpha_1 \sigma_{t-1}^2 + \alpha_2 \varepsilon_{t-1}^2, \tag{3}$$

where $\sigma_t$ is the volatility of target time series and $\varepsilon_t$ is the residual term from the return prediction equation, which is:

$$r_t = \phi + \varepsilon_t, \tag{4}$$

where $\phi$ is the conditional mean, and $\varepsilon_t \sim N(0, \sigma_t^2)$.

Because of the data availability and time series convergence issues, Fig 8 includes six markets' conditional volatilities predicted by GARCH model. It can be seen that the key volatility clustering period was during 2015-2016, which was the stock market crash in China. Our volatility spillover result can be also substantiated by volatility clustering phenomenon. Since high volatilities were concentrated in similar periods, it is thereby arguable that risk might be propagated across different markets during those periods.

## 4.2 Robustness check

In order to illustrate the robustness of our results, we apply both inverse roots of the characteristic AR polynomial test and Kendall's tau covariance analysis in this subsection. Fig 9 presents the test result for inverse roots of the characteristic AR polynomial. This test aims to verify the stability and robustness of our VAR model results. It has been documented that our established VAR model is stable if all roots lie inside the unit circle (Lutkepohl [50]; Tuaneh [51]). From Fig 9, it is clear that all roots of our established VAR model lie inside the unit circle, which indicates that our results are stable and thus robust.

For the robustness purpose, we further adopt the Kendall's tau covariance analysis to estimate the correlation for both volatility and illiquidity measure in different commodity markets. From Tables 7 and 8, it is observable that results are similar with results in Table 3.1 and 3.2. For both illiquidity and volatility correlations for commodity and stock markets, the signs for both positive and negative correlations are nearly the same in Tables 7 and 8 as in Table 3.1 and 3.2. It is thereby arguable that our results are robust.

## 5 Conclusions

To conclude, this paper studies the impact persistence of volatility spillover and illiquidity spillover from the stock market to commodity markets. We use the volatility and illiquidity of the CSI 300 Index to proxy the stock market, and we employ the VAR model to accommodate the impact persistence of volatility spillover and illiquidity spillover. We reveal that there is a cross-market spillover of volatility and illiquidity among Chinese financial markets. We also demonstrate that there is a strong interaction between volatility and illiquidity in financial markets. Finally, we demonstrate that different commodity markets have different responsiveness to stock market shocks. In particular, we discover that copper market volatility shows heavy responsiveness to stock market illiquidity shocks and that the shock impact is persistent, while the illiquidity of the copper market is unresponsive to stock market volatility shocks. On the other hand, corn and soybean market volatilities are insensitive to stock market illiquidity shocks, and the shock impact is persistent, while the liquidities of the corn and soybean

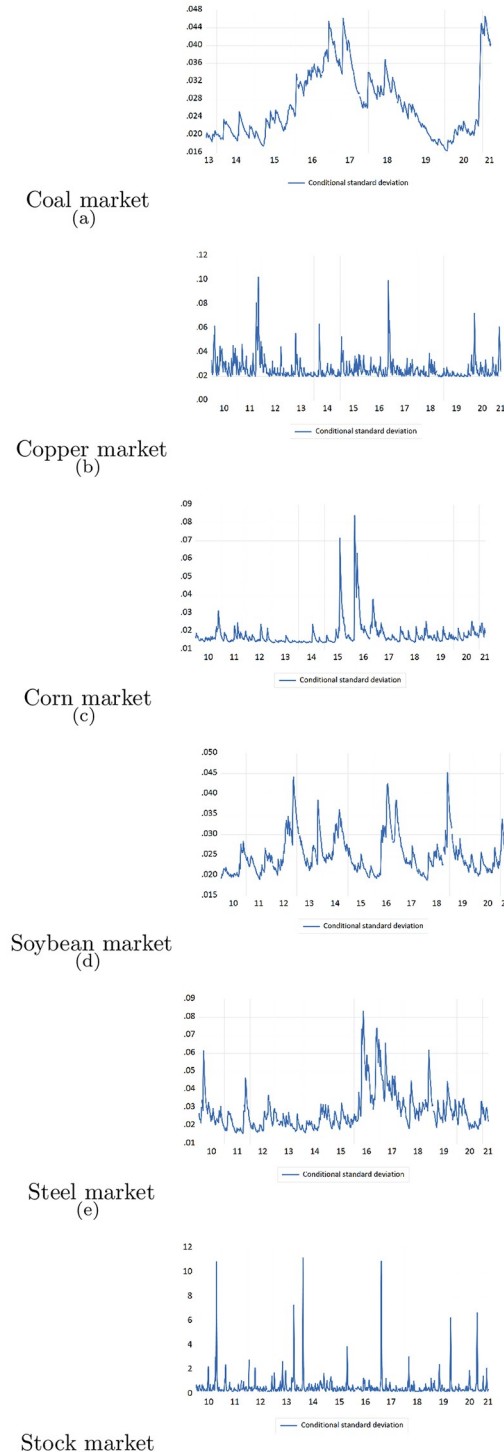

Coal market
(a)

Copper market
(b)

Corn market
(c)

Soybean market
(d)

Steel market
(e)

Stock market
(f)

**Fig 8. Volatility clustering for commodity markets and stock market based on GARCH model volatility predictions.**

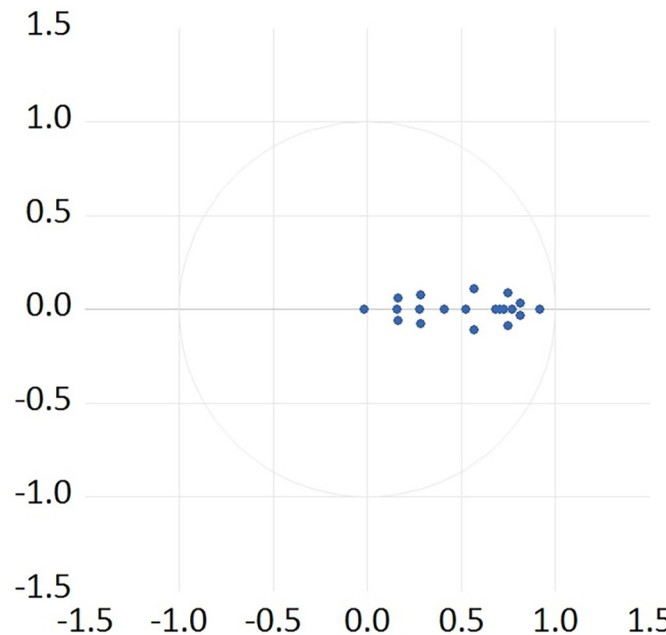

**Fig 9. Inverse roots of the characteristic AR polynomial for our established VAR model.**

**Table 7. Kendall's tau covariance analysis of the illiquidity measurement.** The table presents the correlation matrix of the time-series of illiquidity measures. Illiquidity is measured using the Amihud measure for each market. The sample runs from January 1, 2010 to March 22, 2021.

| | $L_t^{coa}$ | $L_t^{cop}$ | $L_t^{cor}$ | $L_t^{gol}$ | $L_t^{met}$ | $L_t^{oil}$ | $L_t^{soy}$ | $L_t^{ste}$ | $L_t^{sug}$ | $L_t^{csi}$ |
|---|---|---|---|---|---|---|---|---|---|---|
| $L_t^{coa}$ | 1.0000 | | | | | | | | | |
| | — | | | | | | | | | |
| $L_t^{cop}$ | 0.0398 | 1.0000 | | | | | | | | |
| | 0.1709 | — | | | | | | | | |
| $L_t^{cor}$ | 0.0024 | -0.0246 | 1.0000 | | | | | | | |
| | 0.9348 | 0.3775 | — | | | | | | | |
| $L_t^{gol}$ | -0.3157 | 0.0716 | 0.1330 | 1.0000 | | | | | | |
| | 0.0000 | 0.0103 | 0.0000 | — | | | | | | |
| $L_t^{met}$ | 0.0646 | -0.1042 | 0.0128 | -0.0564 | 1.0000 | | | | | |
| | 0.0279 | 0.0002 | 0.6508 | 0.0456 | — | | | | | |
| $L_t^{oil}$ | 0.0093 | -0.0492 | 0.0649 | -0.0447 | -0.0789 | 1.0000 | | | | |
| | 0.7515 | 0.0826 | 0.0220 | 0.1143 | 0.0059 | — | | | | |
| $L_t^{soy}$ | -0.0793 | 0.0804 | 0.1210 | 0.1148 | -0.0603 | -0.0556 | 1.0000 | | | |
| | 0.0063 | 0.0039 | 0.0000 | 0.0000 | 0.0327 | 0.0493 | — | | | |
| $L_t^{ste}$ | -0.0279 | 0.1213 | -0.0029 | 0.1138 | -0.0168 | -0.0194 | 0.1200 | 1.0000 | | |
| | 0.3366 | 0.0000 | 0.9180 | 0.0000 | 0.5520 | 0.4941 | 0.0000 | — | | |
| $L_t^{sug}$ | -0.0898 | 0.0833 | 0.0083 | -0.0378 | 0.1216 | -0.1042 | 0.0492 | 0.0333 | 1.0000 | |
| | 0.0020 | 0.0028 | 0.7664 | 0.1749 | 0.0000 | 0.0002 | 0.0778 | 0.2330 | — | |
| $L_t^{csi}$ | -0.0257 | 0.0278 | 0.0070 | -0.0350 | 0.0267 | -0.0196 | 0.0597 | -0.0054 | 0.0091 | 1.0000 |
| | 0.3760 | 0.3192 | 0.8022 | 0.2095 | 0.3441 | 0.4890 | 0.0324 | 0.8465 | 0.7431 | — |

**Table 8. Kendall's tau covariance analysis of the volatility in commodity markets.** The table presents the correlation matrix of the time-series of volatility. The sample runs from January 1, 2010 to March 22, 2021.

| | $V_t^{coa}$ | $V_t^{cop}$ | $V_t^{cor}$ | $V_t^{gol}$ | $V_t^{met}$ | $V_t^{oil}$ | $V_t^{soy}$ | $V_t^{ste}$ | $V_t^{sug}$ | $V_t^{csi}$ |
|---|---|---|---|---|---|---|---|---|---|---|
| $V_t^{coa}$ | 1.0000 | | | | | | | | | |
| | — | | | | | | | | | |
| $V_t^{cop}$ | -0.0667 | 1.0000 | | | | | | | | |
| | 0.0216 | — | | | | | | | | |
| $V_t^{cor}$ | 0.1733 | 0.1151 | 1.0000 | | | | | | | |
| | 0.0000 | 0.0000 | — | | | | | | | |
| $V_t^{gol}$ | -0.2080 | 0.2107 | 0.0114 | 1.0000 | | | | | | |
| | 0.0000 | 0.0000 | 0.6840 | — | | | | | | |
| $V_t^{met}$ | 0.4214 | -0.0536 | 0.1105 | -0.0861 | 1.0000 | | | | | |
| | 0.0000 | 0.0574 | 0.0001 | 0.0023 | — | | | | | |
| $V_t^{oil}$ | 0.1712 | 0.0685 | 0.1548 | 0.1022 | 0.2127 | 1.0000 | | | | |
| | 0.0000 | 0.0144 | 0.0000 | 0.0003 | 0.0000 | — | | | | |
| $V_t^{soy}$ | 0.0624 | -0.0214 | 0.0516 | -0.0049 | 0.0767 | 0.0422 | 1.0000 | | | |
| | 0.0315 | 0.4438 | 0.0641 | 0.8596 | 0.0066 | 0.1318 | — | | | |
| $V_t^{ste}$ | 0.2520 | 0.0998 | 0.1512 | -0.0980 | 0.1730 | 0.0883 | 0.1414 | 1.0000 | | |
| | 0.0000 | 0.0003 | 0.0000 | 0.0004 | 0.0000 | 0.0016 | 0.0000 | — | | |
| $V_t^{sug}$ | -0.0885 | 0.1598 | 0.0284 | 0.0804 | -0.0213 | 0.0532 | 0.0136 | 0.0185 | 1.0000 | |
| | 0.0023 | 0.0000 | 0.3085 | 0.0039 | 0.4513 | 0.0577 | 0.6267 | 0.5083 | — | |
| $V_t^{csi}$ | -0.0457 | -0.0360 | -0.1189 | 0.0366 | -0.0307 | -0.0023 | 0.0082 | -0.0369 | -0.0574 | 1.0000 |
| | 0.1157 | 0.1962 | 0.0000 | 0.1895 | 0.2759 | 0.9349 | 0.7676 | 0.1856 | 0.0395 | — |

markets vary largely with the stock market volatility shock. Other than the self-dependent market, we label two different types of commodity markets. One type of commodity market, such as agricultural markets, is dominated by investors, and the other commodity market, such as the copper market, is dominated by hedgers. Investors are concerned about the policy impact, so agricultural markets respond to stock market volatility. On the other hand, copper market volatility is strongly influenced by stock market illiquidity, as hedgers are sensitive to market illiquidity. Using the DY spillover index, we also illuminate that the spillover illiquidity spillover might precede volatility spillover for most commodity markets, which can serve as a signal for policy makers to stabilize financial markets in advance.

## Author Contributions

**Conceptualization:** Shusheng Ding.

**Data curation:** Zhipan Yuan.

**Formal analysis:** Zhipan Yuan.

**Methodology:** Shusheng Ding.

**Project administration:** Tianxiang Cui.

**Resources:** Zheng Lu.

**Supervision:** Tianxiang Cui.

**Validation:** Xihan Xiong, Zheng Lu.

**Writing – original draft:** Shusheng Ding, Fan Chen.

Writing – review & editing: Shusheng Ding.

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
