## [Decision Letter · Decision Letter 0]

16 Jul 2021

PONE-D-21-17518

Impact Persistence of Stock Market Risks on Commodity Market Risks

PLOS ONE

Dear Dr. Cui,

Thank you for submitting your manuscript to PLOS ONE. After careful consideration, we feel that it has merit but does not fully meet PLOS ONE’s publication criteria as it currently stands. Therefore, we invite you to submit a revised version of the manuscript that addresses the points raised during the review process.

We look forward to receiving your revised manuscript.

Kind regards,

Junhuan Zhang, PhD

Academic Editor

PLOS ONE

Journal Requirements:

Reviewers' comments:

Reviewer's Responses to Questions

**Comments to the Author**

1. Is the manuscript technically sound, and do the data support the conclusions?

Reviewer #1: Yes

Reviewer #2: Yes

2. Has the statistical analysis been performed appropriately and rigorously? 

Reviewer #1: Yes

Reviewer #2: Yes

3. Have the authors made all data underlying the findings in their manuscript fully available?

Reviewer #1: No

Reviewer #2: No

4. Is the manuscript presented in an intelligible fashion and written in standard English?

Reviewer #1: Yes

Reviewer #2: Yes

5. Review Comments to the Author

Reviewer #1: This paper analyses the impact persistence of volatility spillover and illiquidity spillover from the stock market to commodity markets for the case of China. This manuscript addresses an interesting question. However some revisions should be performed in order to elevate the overall quality of the paper. I present my suggestions and comments below.

1. I would suggest to revise the title, emphasizing the geography (China) of your paper.

2. I would suggest to substantially rewrite the Abstract in order to clearly emphasize what exactly is the main contribution of this paper.

Please be more concrete and assertive while writing the Abstract. For example, how the first two sentences of the Abstract (see below) help the reader to comprehend what is the paper about?:

<<the a="" among="" are="" been="" burgeoning="" emphasized:="" financial="" fluctuation="" has="" in="" investigated="" market="" markets="" number="" of="" return="" risk="" risks="" spillover="" studies.="" the="" two="" types="">>

This statement generates more questions than answers. For example, why are just two types of risks (volatility and liquidity) emphasized in the literature? What is about the credit risk, reputation risk, operational risk, etc.?

Then you continue: <<against backdrop="" this="">>. What exactly is this backdrop you are talking about? In my view, in the first two sentences of the Abstract you have not managed to duly explain this point.

It is not clear what do you mean under the 10 periods, during which the stock market volatility shock persists in the corn and soybean markets.

I would suggest indicating the time frame(s) of your empirical study in the Abstract.

3. The literature survey must be considerably improved in order to position the paper among the related state of the art. I suggest to include a separate section providing a thorough literature review, focusing on the most recent studies. Authors may consider to address the following papers among many others:

Ahmed, A.D., Huo, R. (2021), Volatility transmissions across international oil market, commodity futures and stock markets: Empirical evidence from China. Energy Economics, 93, 104741, https://doi.org/10.1016/j.eneco.2020.104741.

Frömmel, M., Han, X., Li, Y., Vigne, S. (2021), Low liquidity beta anomaly in China. Emerging Markets Review, 100832, https://doi.org/10.1016/j.ememar.2021.100832.

Umar, Z., Gubareva, M., and Teplova, T. (2021), The impact of Covid-19 on commodity markets volatility: Analyzing time-frequency relations between commodity prices and coronavirus panic levels. Resources Policy, 73, 102164. https://doi.org/10.1016/j.resourpol.2021.102164

Umar, Z., Gubareva, M., Naeem, M., and Akhter, A. (2021), Return and volatility transmission between oil price shocks and agricultural commodities. PLOS ONE. 16(2): e0246886. https://doi.org/10.1371/journal.pone.0246886.

Zhang, Y., Ding, S. (2021), Liquidity effects on price and return co-movements in commodity futures markets. International Review of Financial Analysis, 76, 101796, https://doi.org/10.1016/j.irfa.2021.101796.

4. While describing on page 2 (lines 40-45) the main distinguishing features of your research, you abruptly continue with the following text:

<<furthermore, advanced="" also="" comparable="" economies="" in="" is="" our="" research="" risk="" spillover="" study="" the="" with="">>

Please, use the “bridge” phrase to prepare the reader for changing of the direction of your thought from presenting distinguishing features of your research to describing the similarities of your study with the already published research papers.

Moreover, I suggest you to explain in what extent your paper is comparable to the paper by Andrikopoulos et al., 2014, and other more recent papers on this topic.

5. Please explain what is the advantage of the VAR model in the context of your research. Why don’t you employ other alternatives, such as Markov switching model and/ or multiplicative error model (MEM), among others?

6. Please explain what is the advantage of the Amihud measure of liquidity? Why don’t you utilize other alternatives? Authors may address the following recent studies on liquidity measures, while addressing this point:

Díaz, A. and Escribano, A. (2020), Measuring the multi-faceted dimension of liquidity in financial markets: A literature review. Research in International Business and Finance, 51, 101079, https://doi.org/10.1016/j.ribaf.2019.101079.

Cakici, N. and Zaremba, A. (2021), Liquidity and the cross-section of international stock returns. Journal of Banking & Finance, 127, 106123, https://doi.org/10.1016/j.jbankfin.2021.106123.

Gubareva, M. (2021), Covid-19 and high yield emerging market bonds: insights for liquidity risk management. Risk Management. https://doi.org/10.1057/s41283-021-00074-7

Kang, W. and Zhang, H. (2014), Measuring liquidity in emerging markets, Pacific-Basin Finance Journal, 27, pp 49-71, https://doi.org/10.1016/j.pacfin.2014.02.001.

7. Are the results of your paper are robust? I suggest to address the robustness of the results.

8. While the results are quite interesting, there is no thorough discussion of the results. Authors should provide economic interpretation of the obtained results and explain what are the implications of this study, how this study could be used by practitioners, by policy-makers etc.. I suggest to introduce the additional section to address this point.

Summarizing, I found the subject of this research to be interesting. However, Authors need to elevate the quality of this paper to be eligible for publishing in PLoS One in accordance with the suggestions above.

I thank the Authors for the opportunity to consider this manuscript.

Sincerely,

Reviewer</furthermore,></against></the>

Reviewer #2: The manuscript is interesting and easy to read. However, I would like to ask the author(s) to write "volatility" and "liquidity" consistently, whether volatility or liquidity will be written/explained first (form Abstract to Conclusions). Please read the Abstract, for example. I suggest the author(s) to use several volatility proxies and please write the corresponding formulas explicitly. Also, please involve volatility clustering and/or leverage effect since this deals with return-volatility relationship. In addition, it would be better to not only use Pearson's correlation but also Kendall's tau.

6. PLOS authors have the option to publish the peer review history of their article (what does this mean?). If published, this will include your full peer review and any attached files.

Reviewer #1: No

Reviewer #2: No

---

## [Author Response · Author response to Decision Letter 0]

9 Aug 2021

(Also see attached)

Response to Reviewers’ Comments

1. I would suggest to revise the title, emphasizing the geography (China) of your paper.

Response: Thank you for your comment, we have changed the title. 

2. I would suggest to substantially rewrite the Abstract in order to clearly emphasize what exactly is the main contribution of this paper. Please be more concrete and assertive while writing the Abstract. For example, how the first two sentences of the Abstract (see below) help the reader to comprehend what is the paper about?: <> This statement generates more questions than answers. For example, why are just two types of risks (volatility and liquidity) emphasized in the literature? What is about the credit risk, reputation risk, operational risk, etc.? Then you continue: <>. What exactly is this backdrop you are talking about? In my view, in the first two sentences of the Abstract you have not managed to duly explain this point. It is not clear what do you mean under the 10 periods, during which the stock market volatility shock persists in the corn and soybean markets. I would suggest indicating the time frame(s) of your empirical study in the Abstract. 

Response: Thank you for your comment, we have revised our abstract and we have included time frame(s) and research implications. Because of the misleading for the first two sentences, we have now deleted the two sentences. Moreover, we have emphasized our contribution that our study reveals that different commodity markets have different responsiveness to stock market shocks, which indicates the stock market risk impact persistence in Chinese commodity markets. This result can help policymakers to understand the policy propagation effect according to this risk spillover channel and risk impact persistence mechanism in China.

3. The literature survey must be considerably improved in order to position the paper among the related state of the art. I suggest to include a separate section providing a thorough literature review, focusing on the most recent studies. Authors may consider to address the following papers among many others: 

1. Ahmed, A.D., Huo, R. (2021), Volatility transmissions across international oil market, commodity futures and stock markets: Empirical evidence from China. Energy Economics, 93, 104741, https://doi.org/10.1016/j.eneco.2020.104741. 

2. Frömmel, M., Han, X., Li, Y., Vigne, S. (2021), Low liquidity beta anomaly in China. Emerging Markets Review, 100832, https://doi.org/10.1016/j.ememar.2021.100832. 

3. Umar, Z., Gubareva, M., and Teplova, T. (2021), The impact of Covid-19 on commodity markets volatility: Analyzing time-frequency relations between commodity prices and coronavirus panic levels. Resources Policy, 73, 102164. https://doi.org/10.1016/j.resourpol.2021.102164

4. Umar, Z., Gubareva, M., Naeem, M., and Akhter, A. (2021), Return and volatility transmission between oil price shocks and agricultural commodities. PLOS ONE. 16(2): e0246886. https://doi.org/10.1371/journal.pone.0246886.

5. Zhang, Y., Ding, S. (2021), Liquidity effects on price and return co-movements in commodity futures markets. International Review of Financial Analysis, 76, 101796, https://doi.org/10.1016/j.irfa.2021.101796. 

Response: Thank you for your comment, we have added a separate section focusing on the most recent studies. This section includes all the papers mentioned above. 

4. While describing on page 2 (lines 40-45) the main distinguishing features of your research, you abruptly continue with the following text: 

Please, use the “bridge” phrase to prepare the reader for changing of the direction of your thought from presenting distinguishing features of your research to describing the similarities of your study with the already published research papers. Moreover, I suggest you to explain in what extent your paper is comparable to the paper by Andrikopoulos et al., 2014, and other more recent papers on this topic. 

Response: Thank you for your comment, we have revised this part, which makes our paper comparable with existing study. Specifically, our study differentiates from existing studies from two aspects. 

Firstly, extant research focuses on the risk spillover between commodity markets (see Ji et al., 2018; Shahzad et al., 2018; Mensi et al., 2021). Those doctrines mainly adopt volatility as the risk measure to explore the spillover effect between commodity markets. In complementary, our research not only uses volatility but also the market illiquidity risk in our model to broaden the investigation of risk spillover among financial markets. 

More importantly, we employ the VAR (Vector Auto regression) model to analyze the commodity market sensitivity towards the shocks of stock market risk and illiquidity as the commodity market responsiveness. As in Andrikopoulos et al. (2014)’s study, their focus is on stock markets risk spillover in different countries. They unveil Granger causal relationship between risk, return and illiquidity across G7 stock market and they deliver the research implications regarding the international portfolio diversification. In contrast, our focal point rests on the interaction between commodity market and stock market and also, our data sample concentrates in Chinese markets instead of G7 stock markets.

5. Please explain what is the advantage of the VAR model in the context of your research. Why don’t you employ other alternatives, such as Markov switching model and/ or multiplicative error model (MEM), among others? 

Response: Thank you for your comment, the advantage of the VAR model is as follows. The vector autoregression (VAR) model is one of the most successful, flexible, and easy to use models for the analysis of multivariate time series. It is a natural extension of the univariate autoregressive model to dynamic multivariate time series. Our paper is dealing with multiple time series from Chinese commodity markets, which fits the VAR model usage well.

Furthermore, the VAR model has proven to be especially useful for describing the dynamic behavior of financial time series and for forecasting. It often provides superior forecasts to those from univariate time series models and elaborate theory-based simultaneous equations models. Therefore, our results can provide useful insights for commodity forecasting by employing the VAR model. 

The Markov switching model is one of the most popular nonlinear time series models in the literature. This model involves multiple structures (equations) that can characterize the time series behaviors in different regimes. All our time series have passed the unit root test (see Table 2), which indicates that all our variables are stationary and may not vary among different regimes based on nonlinear models. 

In addition, The Multiplicative Error Model (MEM) is borne of an extension of the popular GARCH approach for modeling and forecasting conditional volatility of asset returns. Our study focuses on the risk spillover effect, not on forecasting conditional volatility. As a result, using MEM may deviate from our initial research purpose. 

6. Please explain what is the advantage of the Amihud measure of liquidity? Why don’t you utilize other alternatives? Authors may address the following recent studies on liquidity measures, while addressing this point: 

Díaz, A. and Escribano, A. (2020), Measuring the multi-faceted dimension of liquidity in financial markets: A literature review. Research in International Business and Finance, 51, 101079, https://doi.org/10.1016/j.ribaf.2019.101079. 

Cakici, N. and Zaremba, A. (2021), Liquidity and the cross-section of international stock returns. Journal of Banking & Finance, 127, 106123, https://doi.org/10.1016/j.jbankfin.2021.106123. 

Gubareva, M. (2021), Covid-19 and high yield emerging market bonds: insights for liquidity risk management. Risk Management. https://doi.org/10.1057/s41283-021- 00074-7 

Kang, W. and Zhang, H. (2014), Measuring liquidity in emerging markets, Pacific- Basin Finance Journal, 27, pp 49-71, https://doi.org/10.1016/j.pacfin.2014.02.001. 

Response: Thank you for your comment, Marshall et al. (2012) test a large number of liquidity proxies for 19 commodities. They find that the Amihud liquidity proxy has the maximal correction ratio among all proxies and they strongly recommend researchers to use this proxy when modeling commodity liquidity. Kang and Zhang (2014) reveal that the adjusted Amihud measure provides helpful insights for emerging financial market liquidity. Díaz and Escribano (2020) show that Amihud measure is a three dimensions of liquidity measure, including depth, immediacy and tightness. As a result, we use the proxy mentioned in Amihud (2002) to measure asset illiquidity for this study. 

 In fact, the results for other alternatives like Roll’s measure and Liu’s measure for liquidity could be quite similar in our VAR system. More importantly, our paper aims to demonstrate the risk spillover among financial markets, not to compare results among different liquidity measures. Including other measures may deviate from our initial research purpose. 

7. Are the results of your paper are robust? I suggest to address the robustness of the results. 

Response: Thank you for your comment, we have used Kendall's tau to check the robustness of our correlation table and we use inverse roots of AR characteristic polynomial method to check the stability and robustness of our VAR model, presenting in Table 7 and Table 8, as well as in Fig 9 respectively.

 Nevertheless, VAR system is a successful and stable model system, which is based on Granger causality. As a result, the result for VAR model is usually stable and would not vary with variables and models. 

8. While the results are quite interesting, there is no thorough discussion of the results. Authors should provide economic interpretation of the obtained results and explain what are the implications of this study, how this study could be used by practitioners, by policy-makers etc.. I suggest to introduce the additional section to address this point. 

Response: Thank you for your comment, we have added the policy implications of this paper.

9. I would like to ask the author(s) to write "volatility" and "liquidity" consistently, whether volatility or liquidity will be written/explained first (form Abstract to Conclusions). Please read the Abstract, for example. I suggest the author(s) to use several volatility proxies and please write the corresponding formulas explicitly. Also, please involve volatility clustering and/or leverage effect since this deals with return- volatility relationship. In addition, it would be better to not only use Pearson's correlation but also Kendall's tau. 

Response: Thank you for your comment, we have written "volatility" and "illiquidity" consistently. A simple way for volatility clustering is to consider a long time series, is based on a GARCH (1,1) process. Since the volatility stems from GARCH model is conditional volatility, which differs from our previous volatility and we have written the corresponding formulas for GARCH model explicitly. On the basis, we have included volatility clustering effect in our result, presenting in Fig 8. Finally, we added Kendall's tau in our paper, presenting in Table 7 and Table 8.

---

## [Decision Letter · Decision Letter 1]

18 Oct 2021

Impact Persistence of Stock Market Risks in Commodity Markets: Evidence from China

PONE-D-21-17518R1

Dear Dr. Cui,

We’re pleased to inform you that your manuscript has been judged scientifically suitable for publication and will be formally accepted for publication once it meets all outstanding technical requirements.

Kind regards,

Junhuan Zhang, PhD

Academic Editor

PLOS ONE

Additional Editor Comments (optional):

Reviewers' comments:

Reviewer's Responses to Questions

**Comments to the Author**

1. If the authors have adequately addressed your comments raised in a previous round of review and you feel that this manuscript is now acceptable for publication, you may indicate that here to bypass the “Comments to the Author” section, enter your conflict of interest statement in the “Confidential to Editor” section, and submit your "Accept" recommendation.

Reviewer #1: All comments have been addressed

Reviewer #2: All comments have been addressed

2. Is the manuscript technically sound, and do the data support the conclusions?

Reviewer #1: Yes

Reviewer #2: Yes

3. Has the statistical analysis been performed appropriately and rigorously? 

Reviewer #1: Yes

Reviewer #2: Yes

4. Have the authors made all data underlying the findings in their manuscript fully available?

Reviewer #1: (No Response)

Reviewer #2: Yes

5. Is the manuscript presented in an intelligible fashion and written in standard English?

Reviewer #1: (No Response)

Reviewer #2: Yes

6. Review Comments to the Author

Reviewer #1: This paper deals with an interesting and timely topic. The methodology adopted is suitable and adequate for the topic under investigation. The section of empirical findings is well written and presents interesting and convincing discussions. The robustness checks are provided. Conclusion section of the paper is justified by the results. All my recommendations and comments were considered by the Authors and included in the revised

manuscript.

I thank the Authors for the opportunity to consider their work.

Sincerely,

Reviewer

Reviewer #2: The authors have addressed all my comments. Thank you. It would be better to state/remark for other possibility of modeling with extension of GARCH.

7. PLOS authors have the option to publish the peer review history of their article (what does this mean?). If published, this will include your full peer review and any attached files.

Reviewer #1: No

Reviewer #2: No

---

## [Editor Report · Acceptance letter]

28 Oct 2021

PONE-D-21-17518R1 

Impact Persistence of Stock Market Risks in Commodity Markets: Evidence from China 

Dear Dr. Cui:

I'm pleased to inform you that your manuscript has been deemed suitable for publication in PLOS ONE. Congratulations! Your manuscript is now with our production department. 

Kind regards, 

on behalf of

Dr. Junhuan Zhang 

Academic Editor

PLOS ONE